# Climate Change Impacts on Grassland Vigour in Northern Portugal

Oiliam Stolarski [ID], João A. Santos [ID], André Fonseca [ID], Chenyao Yang [ID], Henrique Trindade [ID]
and Helder Fraga *[ID]

Centre for the Research and Technology of Agri-Environmental and Biological Sciences (CITAB), Institute for Innovation, Capacity Building and Sustainability of Agri-Food Production (Inov4Agro), Universidade de Trás-os-Montes e Alto Douro (UTAD), 5000-801 Vila Real, Portugal; jsantos@utad.pt (J.A.S.); andref@utad.pt (A.F.); cyang@utad.pt (C.Y.); htrindad@utad.pt (H.T.)
*  Correspondence: hfraga@utad.pt

**Abstract:** Grasslands are key elements of the global agricultural system, covering around two-thirds of all agricultural areas and playing an important role in biodiversity conservation, food security, and balancing the carbon cycle. Climate change is a growing challenge for the agricultural sector and may threaten grasslands. To address these challenges, it is vital to conduct in-depth climate studies to understand the vulnerability of grasslands. In this study, machine learning was used to build an advanced model able to evaluate the future impact of climate change on grassland vigour. The objective was to identify the most vulnerable grassland areas, analyse the interaction between climate and grassland performance, and outline management strategies against the detrimental implications of climate change. A Random Forest (RF) regression was used to model the Normalised Difference Vegetation Index (NDVI) using the Standardised Precipitation-Evapotranspiration Index (SPEI). The model explained 76% of the NDVI variability. The foremost significant predictors of grassland vigour are the SPEI with temporal lags of 1, 4, and 12 months. These findings suggest that the vegetative status of grasslands exhibits high sensitivity to short-term drought while also being influenced by the memory of past climatic events over longer periods. Future projections indicate an overall reduction in grassland vigour, mostly in RCP8.5. The results indicate that negative effects will be more pronounced in mountainous regions, which currently host the most vigorous grasslands. Dry lowlands in the north should continue to have the lowest vigour in the future. A substantial reduction in vigour is expected in autumn, with an effect on grassland phenology. The development of grasslands in winter, favoured by increasing temperatures and precipitation, can advance the harvesting of grassland (cutting) and the grazing of livestock. To ensure that vigour is maintained in less favourable zones, adaptation measures will be needed, as well as more efficient management of highlands to provide an adequate level of production.

**Keywords:** machine learning; random forest; standardised precipitation and evaporation index (SPEI); normalised difference vegetation index (NDVI); climate change; grasslands vigour

## 1. Introduction

Over the last two decades, roughly two-thirds of agricultural land worldwide was used for grasslands [1]. Currently, the global agricultural land area is 4.74 billion hectares, of which 67% is used for permanent meadows and pastures, while 33% is for cropland. China, Australia and the United States of America are the main countries, with about 30% of the world's permanent meadows and pasture areas [1]. Different types of grass and forage terminologies are traditionally distinguished [2,3]. Grasslands can be understood as land devoted to foraging production cutting, grazing/browsing, or used for other agricultural activities, such as renewable energy production [3,4]. The flora of grasslands consists of grasses or grass-like plants, legumes and other forbs, though woody species can also be found in some ecosystems [5], such as in the Mediterranean Montado ecosystem [6].

In the European Union, permanent pastures, which are a type of grassland, cover 35% of the agricultural area [7]. Particularly notable are the grasslands in the Mediterranean Basin (consisting of vast areas of Southern Europe, North Africa and the Middle East). The Mediterranean Basin is a biodiversity hotspot, with many indigenous plant species found in its vast grasslands, which are heavily influenced by long-term management techniques [8]. Grassland-based farming systems are crucial for the Mediterranean regions because they meet the rising demand for animal products, providing smallholders with financial security and producing high-value foodstuffs [5], in addition to several ecosystem services such as wildfires prevention, soil erosion control or biodiversity preservation.

In Portugal, grasslands, typically formed by permanent pasture and spontaneous herbaceous vegetation, have a key level of importance, covering approximately 22% of mainland Portugal, showing an increasing trend in the last decades [9]. These areas provide forage for domestic livestock production, generating products such as milk, meat, and fibre [10,11]. Grasslands are also highly valuable regarding their multiple ecosystem functions, with a significant impact on the global carbon cycle, as they have vast potential for carbon sequestration through the fixation of atmospheric $CO_2$ in plant biomass [12]. In fact, grasslands seem more capable than forests themselves of maintaining carbon sinks [11], especially in locations prone to drought and wildfires [13]. Grasslands can be important drivers of biodiversity, providing valuable habitats for hundreds of species of plants and animals. In addition, grasslands have an impact on ecological processes, ranging from the landscape level (such as pollination and biological control of agricultural pests) to the regional (such as water regulation and purification, erosion prevention, recreation, and cultural values) and global levels (e.g., climate regulation) [10,14,15].

Grassland development depends heavily on climatic conditions, making it particularly vulnerable to climate variability and change [11,16]. As reported by the Intergovernmental Panel on Climate Change (IPCC), surface temperatures are projected to increase significantly by the end of the 21st century [17]. The mean surface warming of the planet is expected to exceed +1.8 °C (RCP4.5; moderate scenario) and may eventually exceed +3.7 °C (RCP8.5; severe scenario) [17]. Furthermore, precipitation patterns are also expected to change considerably, particularly in Southern Europe, where a significant decrease is projected. Another important component of climate change, reported by [18], is the increase in intra- and inter-annual variability. All climate scenarios point to negative effects on the environment, particularly due to the increased frequency of occurrence of extreme weather events, such as heatwaves and droughts [19].

Changes in the physical and chemical composition of plants can result from rising global temperatures and, in particular, drought intensity [20]. Extreme drought can cause tissue senescence, which significantly reduces the overall quality of the forage. Plant maturation accelerates under mild heat stress, but the water content of plant tissues drops, and the amount of water-soluble carbohydrates rises [18]. All these physiological effects can impact the vigour of the grasslands, potentially affecting animal feed imports from out of the farm and the financial stability of producers. Hence, it is of utmost importance to monitor grassland health and intervene in a timely manner. Nonetheless, site-based monitoring is difficult to implement, particularly in large areas, mainly due to the lack of resources.

In recent years, remote sensing technologies have emerged to monitor plant development. Particularly, the advancement of high spatial resolution sensors installed in satellite platforms is becoming a reliable source of information for grassland monitoring [4]. Changes in grassland vigour can be measured using the Normalised Difference Vegetation Index (NDVI) [6,21], which is greatly influenced by drought [22]. NDVI is also used to assess grassland degradation [21], vegetation growth [23], the richness of grassland [24], and pasture quality [6]. While there are other remote sensing indices also linked to vigour, such as the GPP/NPP (Gross/Net Primary Production) or the LAI (Leaf Area Index), these show strong correlations with the NDVI over grasslands [25,26].

Several climatic indices have been developed to assess the extent and severity of dryness [27]. The Standardised Precipitation and Evaporation Index (SPEI) is a water-balance drought index that can be used to assess agricultural water availability for plants and biomass production [28,29]. It is considered one of the most advanced drought indices, partly because it takes into account the relationship between precipitation and evapotranspiration for several monthly timescales (lags). Previous studies have shown that there is a strong relationship between the SPEI and NDVI [30], strengthening the connection between vegetative vigour and drought. This relationship might be crucial to understanding the impact of climate change on pasture vigour. In the literature, there are several works that explore the possible future effects of climate change on pastures [31,32]. However, approaches that seek to relate NDVI and SPEI are scarce [33], particularly over a distant time horizon. Therefore, our study aims to overcome this limitation by using future projections of the SPEI to predict the NDVI over a grassland area. This will indeed enable assumptions on future regional grassland vigour.

To help farmers manage forage supplies in the face of increasing drought conditions and seasonal variability, it is vital to understand more about how climate change is affecting grasslands (Soussana, Klumpp, and Ehrhardt 2014; Liu et al. 2017) [16,22], especially in regions where the livestock sector plays an important socio-economic role. Early notification of the probable negative impacts of climate change, particularly drought-related impacts, on forage production would enable tactical management of grassland and/or livestock decisions, such as the implementation of pasture rotations, adjustment of stocking density, advanced notice of possible needs of earlier supplemental feeding, or acquisition of additional forage resources [28]. In the present study, we applied a machine learning approach to spatiotemporally assess the impacts of future climatic conditions on the vegetative vigour of grasslands in the Côa region, northeastern Portugal. The following questions are addressed more precisely: which grassland areas are most susceptible to climate change? How damaging are the effects of future climate change on grassland vigour? What are the short and long-term management strategies that can possibly be proposed to reduce climate change effects on grasslands?

## 2. Materials and Methods

### 2.1. Study Area

The study area corresponds to the Côa region (Figure 1a,b), which extends about 5.6 thousand km$^2$ across twelve municipalities and covers the entire valley of the Côa River and part of the Douro and Mondego rivers basins at elevations ranging from 95 m (Douro River) to 1285 m (Guarda's municipality mountain ranges) above mean sea level [34]. The Côa region presents a Mediterranean-type climate, with annual mean temperatures of 10.8–16.4 °C and total annual precipitation of 473–1130 mm (Figure 1c,d). June, July and August (Summer) are the warmest and driest months, with a mean temperature of over 22 °C and monthly precipitation of less than 14 mm. On the other hand, Autumn, Winter, and early Spring (from October to April) correspond to the rainiest period (Figure 1e). The southern region shows the highest values of annual precipitation and the lowest average temperatures (higher-elevation area), while the northern region, near the Douro River, is significantly warmer and drier.

### 2.2. Future Vigour Model

The study employs a methodology that integrates spatial reflectance indices and climatic data to construct a vigour model. The model can be used to project the spatial variability of grassland vigour for the Côa region in the context of climate change. Specifically, the machine learning regression model Random Forest (RF) [35] was fitted to the NDVI using SPEI data as predictors. The following sections present a detailed description of the model and the data used.

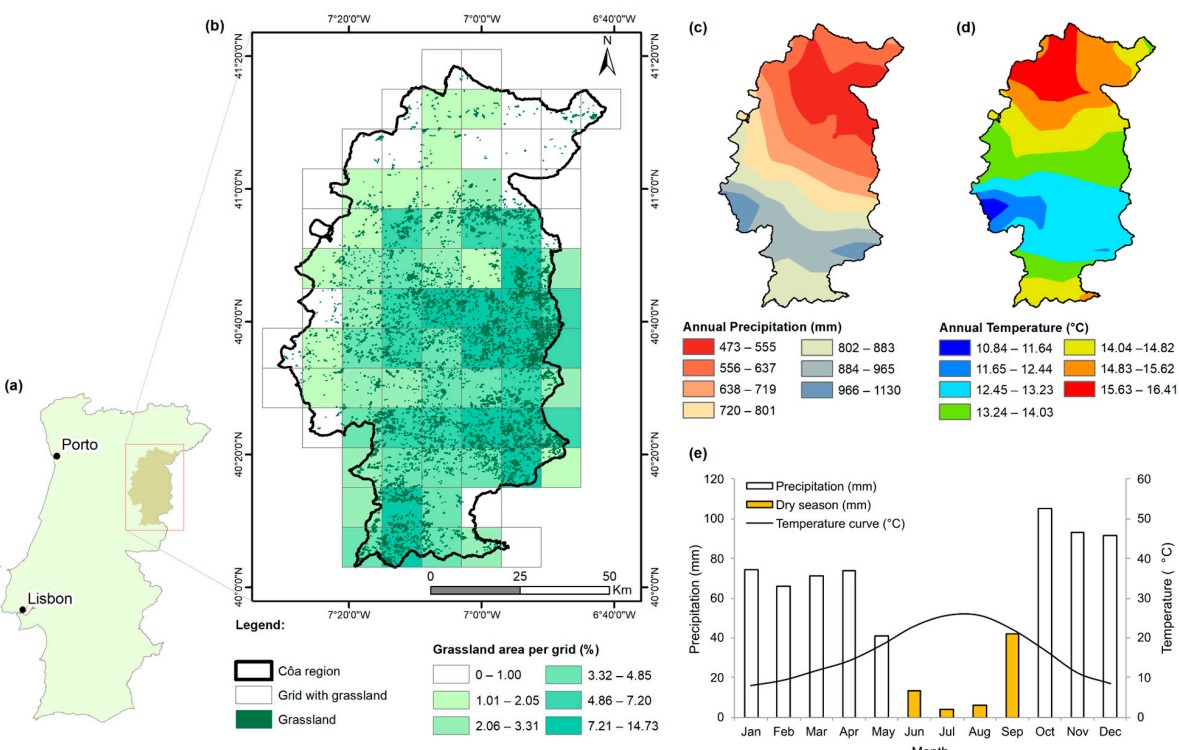

**Figure 1.** Study area presentation: (**a**) Location of the study area in the Côa Region, northeastern Portugal. (**b**) Spatial distribution of the consolidated pasture polygons used in the temporal variability analysis, with the representativeness of the pastures (in % of the land cover area) within each grid box (10 × 10 km) and definition of the grids used in this study. Spatial patterns of (**c**) total annual precipitation and (**d**) mean annual temperature of the Côa region, determined from historical values for the period 2000–2021. (**e**) Gaussen ombrothermic diagram, with monthly precipitation totals (in mm, dry months in orange bars) and mean temperatures (in °C) for the period 2000–2021.

### 2.2.1. Grassland Data

To assess the grassland location in the Côa region, the Portuguese COS ("Carta de Uso e Ocupação do Solo") land cover datasets referring to the years 2007, 2010, 2015, and 2018 were used [36]. Taking into account the COS classification, we considered as grasslands the 3.1.1 and 3.1.2 classes (e.g., COS2018), which correspond to permanent and spontaneous grasslands, respectively. The COS dataset does not include information about species distribution; however, it is known that a large number of autochthonous species can be found herein (Table S1). [37] identified the most common species in this region, including several leguminous species, such as *Ornithopus compressus* and *Trifolium subterraneum*, are found herein. Additionally, *Medicago arabica* is highlighted for its resilience in drier conditions. Among the grasses, *Lolium multiflorum*, *Lolium rigidum*, and *Lolium perenne* (ryegrasses) are prevalent, along with *Dactylis glomerata* and *Phalaris aquatica*. Various species from the genera Bromus, Vulpia, Poa, and Festuca are also significant in these areas, with *Poa pratensis* particularly abundant in meadows, showcasing its persistence.

According to the metadata documentation of the COS, permanent grasslands correspond to areas occupied for 5 years or longer, with essentially herbaceous vegetation, often improved by fertilisation, cropping, seeding, and draining, while spontaneous grasslands are those formed by herbaceous vegetation free of management operations, occupying at least 25% of the surface on which they develop. Given that the COS datasets account for multiple years (from 2007 to 2018), a preliminary spatiotemporal variability analysis of the grasslands was performed to determine the areas for which the land use remained the same during the entire period (overlap COS data), creating a consolidated dataset. This enabled studying the trends in grassland vigour.

### 2.2.2. NDVI Data

In the present study, to assess grassland vigour, NDVI was used. This index, developed by [38], is expressed by the formula: (NIR − Red)/(NIR + Red), where NIR represents the reflectance in the near-infrared spectrum and Red represents the reflectance in the red spectrum. The NDVI was previously used in many studies as a proxy for grassland vigour [6,21]. NDVI data from the MODIS/Terra database [34] were selected to obtain a time series for the Côa Region. More specifically, data from the product MOD13Q1, v6.1, tile H17V04, for the period from 2000 to 2021, were retrieved (23 images per year for a total of 503 images; note that in 2000 the first 3 images are missing). The temporal resolution for the dataset consists of 16-day composites (e.g., day 1, 17, . . ., 353). These composites are produced from maximum daily observations in order to create a nearly cloud-free image. Generally, this dataset contains two images in each month (note that two 16-day images may result in more than a one-month timespan). MOD13Q1 v6.0 was used to complement v6.1 in case of missing data. Monthly data were obtained by averaging NDVI timesteps and attributed to each month of the year between 2000 and 2021 (263 months), and are shown in Figure 2c. Following this procedure, the uncertainties in the dataset were reduced (e.g., unusual values), and no pixel was left without an associated value. Subsequently, a spatiotemporal analysis of the NDVI for each pasture location inside the Côa region was carried out (Figure 1b). As each pasture can contain several NDVI pixels, the mean NDVI value inside each pasture location was computed (Figure 2a) as per the procedure presented in Section 2.2.4.

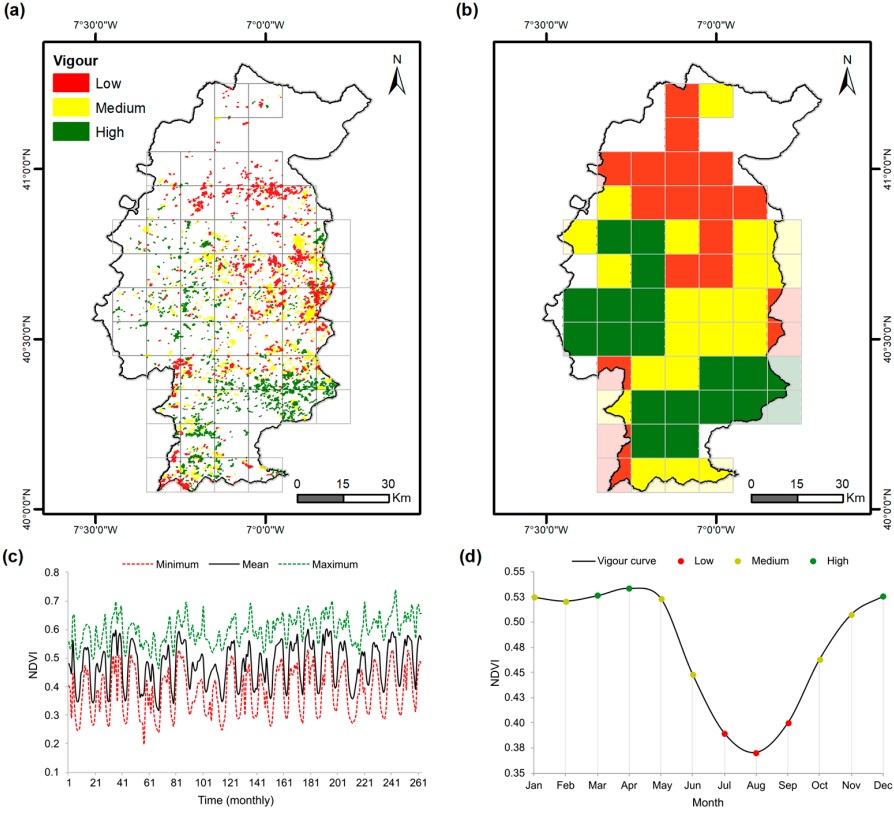

**Figure 2.** Methodological approach: (**a**) Mean monthly NDVI value of the tile for each polygon of consolidated pasture. (**b**) Adjustment of the spatial resolution of the mean NDVI values for consolidated grasslands (250 metres to 10 × 10 km). (**c**) Temporal variability of the NDVI for the whole Côa region, with the minimum, mean, and maximum values plotted. (**d**) The vigour curve of consolidated pasture in the Côa region. Figures (**a**,**b**,**d**) refer to the average NDVI value for all study periods (2000 to 2021).

### 2.2.3. Climatic Data

The SPEI, which was developed by [39], is computed from the difference between precipitation and potential evapotranspiration (PET). SPEI values are understood as the number of standard deviations from the mean conditions and can be positive (humid conditions) or negative (dry conditions). The representation of SPEI values can be undertaken on different time scales SPEI-1, SPEI-2, ..., SPEI-n. As an example, June SPEI-3 would correspond to the number of standard deviations ($+/-$) from the mean for a 3-month period (lag), such as from April to June. This scalability enables the evaluation of the cumulative effects of local weather/climate for a range of time periods [28]. Thus, SPEI is considered to have the potential to express annual grassland productivity as sketched by [29], explaining more than 60% of pasture growth.

The monthly precipitation and temperature values were extracted from two different climatic datasets to compute this index. The calculation was performed for the same period as the historical series of the NDVI, i.e., 263 months starting from February 2000. However, the SPEI requires an additional backlog of data to calculate the first values (e.g., February SPEI-6 requires the previous 6 months). Hence, for 1999–2015, the Iberia01 database [40] was used, which has a spatial resolution of 0.1° (~10 km). The Iberia01 is considered the most reliable climatic dataset for Portugal, though data are only available until 2015. For this reason, for the period 2016–2020, the values were extracted from the E-OBS dataset [41], v.25.0e, with the same resolution [42]. Subsequently, the Iberia01 database, which have more realistic precipitation patterns [43], was used for a bias correction of the E-OBS data based on the delta method [44]. To calculate the PET, the Hargreaves equation was used [45], which is recognised by FAO as an alternative method to the Penman–Monteith formulation [46] since the latter typically requires additional meteorological variables (e.g., radiation, wind speed, and humidity).

### 2.2.4. Data Integration

The present study used different datasets in different formats and spatial resolutions. The grassland locations consisted of vector data at a minimum mapping unit of 1 ha. The NDVI MODIS dataset is comprised of raster data at 250 m spatial resolution. Lastly, the gridded climate data is defined at a 10 km spatial resolution (80 grid cells). The spatial resolution of the abovementioned data was harmonised. Firstly, only the NDVI pixel values inside each grassland polygon were averaged (Figure 2a). Subsequently, for the joint assessment of NDVI grassland data and climate data, the mean NDVI grassland values within each climate grid were averaged. This resulted in a 10 km grid spacing with the mean value of the grassland NDVI. The original resolution of the pasture NDVI, which was 250 metres, was thus adjusted to a 10 × 10 km grid (Figure 2b). To remove outliers in the data (very small grassland polygons in areas dominated by other land cover types), only grassland areas that correspond to a minimum threshold of 1% in 10 km (coarsest of all dataset resolutions) were considered. Hence, some small border areas in the Côa region were left out from this analysis (<100 hectares) (Figure 1b). Therefore, of the 80 cells with consolidated pastures in the Côa region, only 58 grid cells were used. This procedure was carried out for each month separately, considering the 58 grid cells along the historical series. The information was first analysed to verify the temporal evolution of the data set (Figure 2c) and the ability to describe the typical grassland cycle (Figure 2d), which for the target region typically starts in October. The cool winter temperatures interrupt vegetation growth until the temperature increases at the end of February, while pastures start to revive in March. The higher spring temperatures and the moisture accumulated in the soil during the winter will cause the typical Mediterranean spring herbage flush [10].

The value of SPEI-1 through SPEI-12 was computed for each of the 58 grid cells. In this process, it was possible to identify periods when precipitation exceeded evapotranspiration (Figure 3a) or when the scarcity of precipitation led to dryness conditions (Figure 3b). For the study period, humid years (2001, 2003, 2007, 2010, 2011, 2013, 2014, 2016, and 2018) and dry years (2000, 2002, 2004, 2005, 2006, 2008, 2009, 2012, 2015, 2017, 2019, 2020, and

2021) were isolated (Figure 3c,d). The classification used to identify dry events was based on [47], which states that values lower than zero generally indicate drought conditions, but if they exceed the threshold of −1.5 or −2, droughts can then be classified as severe or extreme, respectively.

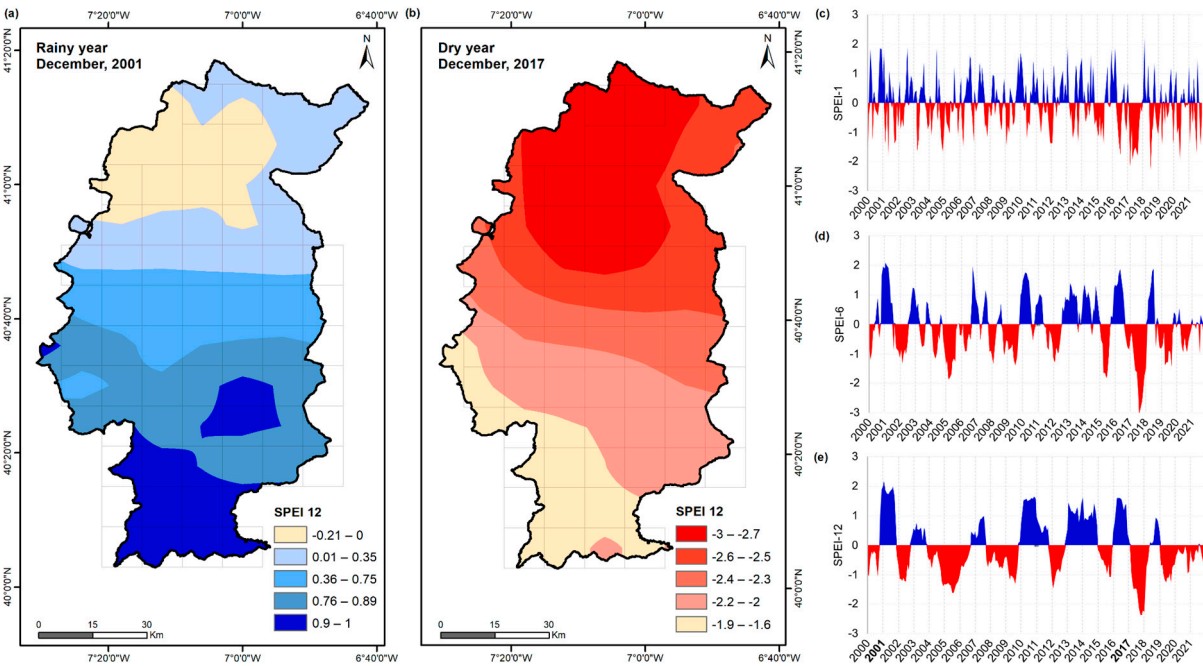

**Figure 3.** Visualisation of the SPEI (Standardised Precipitation-Evapotranspiration Index) for the Côa region: (**a**) Spatial distribution of the SPEI-12 for 2001, a very rainy year. (**b**) Spatial distribution of the SPEI-12 for the year 2017, a very dry year. (**c**) SPEI-1, (**d**) SPEI-6 and (**e**) SPEI-12 time series for the period 2000–2021.

### 2.3. Data Analysis and Modelling

A joint analysis between the monthly SPEI-1 through SPEI-12 with historical NDVI values was carried out to develop a model able to produce estimates of future NDVI-values for the grasslands of the Côa region and, as a consequence, to obtain projections of grassland vigour. As previously mentioned, a machine learning approach, Bootstrap Aggregation (Bagging), was used. This method, also known as the Random Forest (RF) model, uses an ensemble of decision trees for regression [35]. In this model, a random sample of data in a training set is selected with replacement (bootstrapping). After several data samples are generated, each tree is then trained independently (parallelly). The decisions of each tree are then combined to make the final result, effectively dealing with the problem of overfitting (e.g., good and bad models). For the RF model, 15,254 data points were used (263 months × 58 grid cells), and a 10-fold cross-validation was performed. The predictors (features) include monthly SPEI-1, SPEI-2, . . ., and SPEI-12 to predict the monthly NDVI (e.g., NDVI in January is affected by SPEI-1 (SPEI in January), SPEI-2 (SPEI in December-January), and so forth). Several hyperparameter combinations were tested to obtain the best possible RF model (e.g., number of trees and depth), including automatic hyperparameter optimisation (hyperparameter tuning using Bayesian optimisation). Finally, the most important hyperparameter in our testing was the number of trees, which was set at 153 (further increasing this number did not significantly improve the model while requiring much longer data processing time). The modelling approach was performed in two steps: (i) a preliminary model including all SPEI features and (ii) a final model including only the most important features defined in (i).

A preliminary model considered all the predictor variables and obtained $R^2 = 0.93$. The consistency of the information presented by the model was proven by the low correlation

among predictors (Figure 4a), where it was possible to observe a low relationship between the predictor variables of short (SPEI-1, SPEI-2, and SPEI-3) and a long period (SPEI-10, SPEI-11, and SPE-12). The most important features were SPEI-1, SPEI-4, and SPEI-12 (Figure 4b). A new final model was produced using only these predictors, achieving a high determination coefficient, explaining 76% of the NDVI variability.

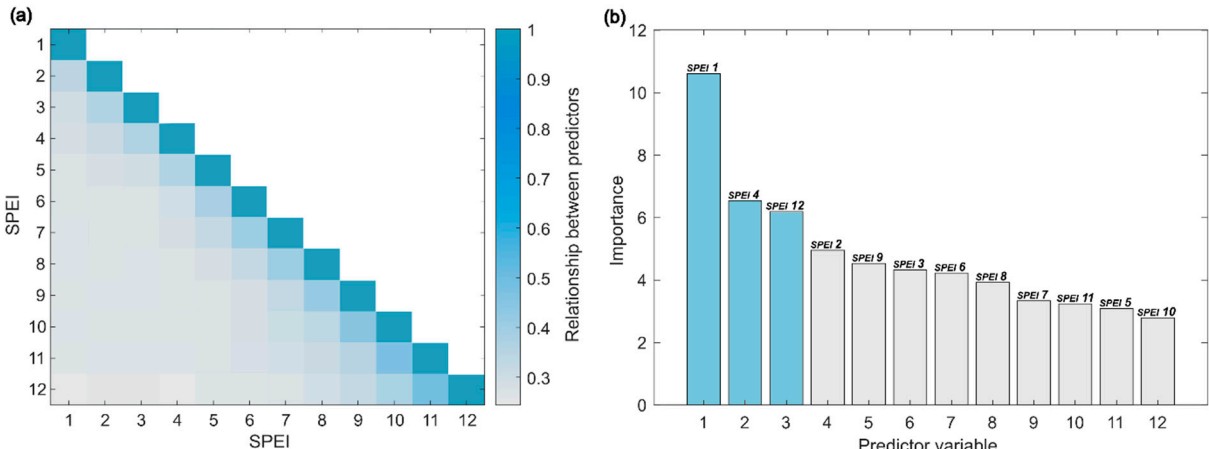

**Figure 4.** Predictor variables selected for Random Forests (RF). (**a**) Predictor association estimates. (**b**) Estimates of feature importance indicating the more relevant predictors for the RF model.

### 2.4. Future Projections

To assess climate change impacts on grassland variability, future climate projections were obtained based on the historical records (2000–2021) of precipitation and temperature and the EURO-CORDEX project [48]. Monthly time series bias-adjusted were obtained using the delta method, under RCP4.5 and RCP8.5 (two anthropogenic radiative forcing scenarios), and for the long-term future period 2071–2100 (Table 1). The delta method has been used in many previous climate change studies [49,50] since it provides a straightforward and transparent approach to estimating climate change impacts, reducing the sensitivity to single-model biases since it is based on multi-model outputs, and is consistent with historical observations since the delta is applied to the observational datasets. The 2071–2100 time period was selected to assess the strongest potential changes in climate. According to these climatic projections, in the Côa region, all seasons are expected to be warmer, with drier spring, summer, and autumn. On the other hand, more precipitation is projected for the winter period. For each scenario, the calculation of the SPEI-1, SPEI-4, and SPEI-12 was carried out and then used as predictor variables to obtain future NDVI values from the final model.

**Table 1.** Reference values from a study by Jacob et al. (2014) for a high spatial resolution dataset projected for 2071–2100. These values were used in this study for fitting the values of monthly precipitation (%) and temperature (°C) deltas for the period 2000–2021, according to the season.

| Season | RCP4.5$_{delta}$ | RCP8.5$_{delta}$ |
|---|---|---|
| Precipitation (%) | | |
| Autumn | −10 | −20 |
| Winter | 10 | 10 |
| Spring | −10 | −20 |
| Summer | −20 | −30 |
| Temperature (°C) | | |
| Autumn | 2.75 | 4.75 |
| Winter | 1.25 | 3.25 |
| Spring | 1.75 | 2.75 |
| Summer | 2.75 | 5.25 |

## 3. Results

### 3.1. Recent-Past Grassland Vigour

The spatial distribution of grasslands for the recent past (2000–2021) is shown in Figure 5a. Lower grassland vigour values were observed in the northern Côa, an area with lower precipitation (Figure 1c), higher temperatures (Figure 1d), and lower elevations (Figure 6). The most vigorous grasslands occur in mountainous regions at elevations above ~700 m (Figure 6), particularly in the municipalities of Sabugal, Guarda, and Celorico da Beira. Areas of medium vigour are frequently found at intermediate elevations, between 600 and 700 m (Figure 6). Currently, the largest grassland areas (Figure 1b) are found in regions associated with high and medium vigour classes (Figure 5a).

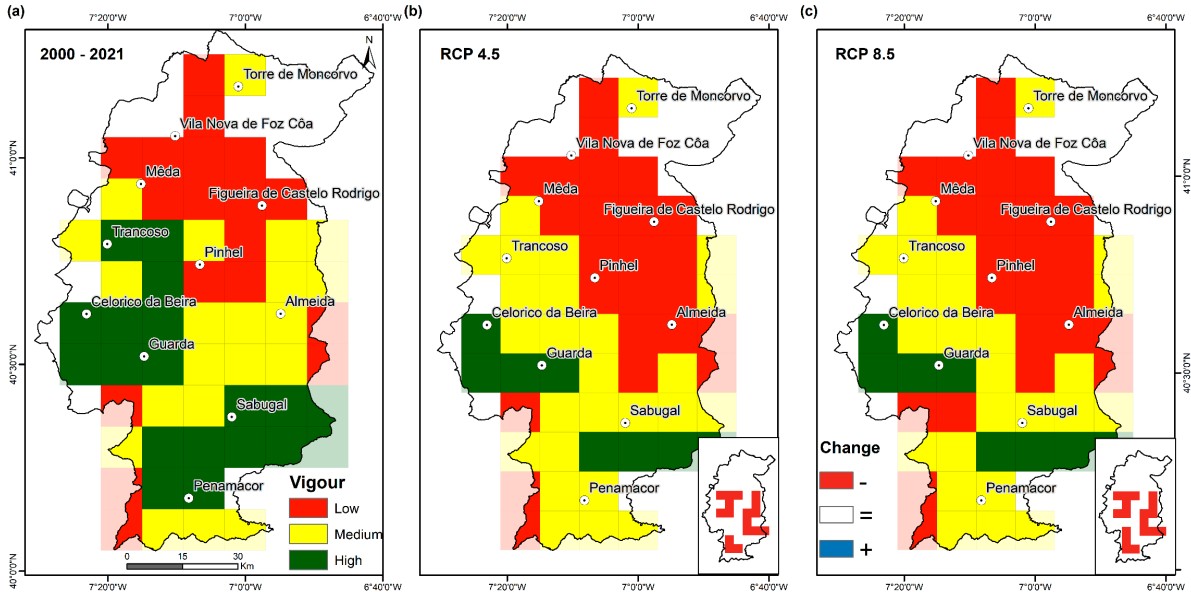

**Figure 5.** Grassland vigour classes (low, medium, and high, based on the recent-past terciles) in the Côa region for the recent-past (2000–2021) and for two anthropogenic radiative forcing scenarios for the long-term future period (2071–2100): present (2000–2021) (**a**), RCP 4.5 and (**b**) and RCP 8.5 (**c**), obtained from the mean of NDVI-values. The relative change is also depicted for each RCP (−: negative, =: no change, +: positive). Ordinal values can be found in Supplementary Figure S2.

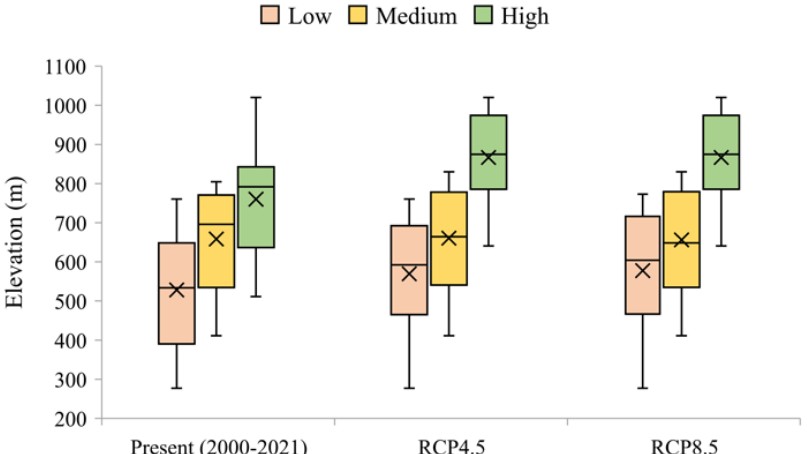

**Figure 6.** Boxplot diagram of grassland vigour classes, arranged by elevation, for the recent past (2000–2021) and future (2071–2100) RCP4.5 and RCP8.5. The vigour classes (low, medium, and high) were defined from the quantiles. The horizontal line within the boxes represents the median, and the × refers to the mean.

### 3.2. Future Changes in Grassland Vigour

Projections for the future (2071–2100) show similar spatial patterns to the recent past period but with a noticeable decreasing trend in vigour. These projections show a southward expansion of the lower vigour areas towards the central-eastern Côa. Medium vigour areas are expected to expand, especially in the southwest or northwest, due to the decrease of high vigour areas. Particularly, the Trancoso region (west) and Penamacor (south), which in the recent past were considered high-vigour areas, are projected to shift to the medium vigour class (Figure 5b,c). To the south of the Guarda municipality, the reduction in vigour is projected to be noticeable, with stronger impacts in RCP8.5 (Figure 5c). In fact, in the more severe scenario, the effects of climate change on grasslands will be generally more visible (Figure S1), with a more pronounced reduction in future NDVI. Nonetheless, the vigour classes (low, medium, and high) are projected to remain similar under both future scenarios (not sufficient to change the vigour class), with the exception of the areas south of Guarda (Figure 5b,c). The areas that are expected to still have high vigour for pastures will be restricted to some parts of Guarda, Celorico da Beira, and some areas north of Penamacor. Nonetheless, the strongest reductions in NDVI are indeed expected for these areas (Figure S1), which also correspond to higher elevations. It should also be noticed that in the current low vigour areas (north), a slight increase in NDVI levels is expected (Figure S1), although also not resulting in a class shift (Figure 5b,c).

In both future scenarios, areas with low vigour grasslands are expected to remain in this class (Figure S2). Conversely, expressive changes are projected in the grasslands defined as medium vigour, changing to low vigour classes, in the order of 33% and 29% for RCP85 and RCP45, respectively (Figure S2). The greatest changes are expected for the high vigour class, where for both RCPs, only 42% are projected to remain in the classes with the high value, while the remaining should shift to the medium vigour class. Grasslands established in the lowlands should remain the less vigorous, while in the highlands (Figure 6), the highest vigour classes still prevail. Nonetheless, according to these projections, there should be an increase in the lower vigour grasslands found at higher altitudes (Figure 6).

### 3.3. Seasonal Changes in Grassland Vigour

The vigour maps for each season, as shown in Figure 7, align with the growth cycle of grasslands. Typically, the pasture cycle commences in autumn, followed by a rapid increase in vigour until winter. It then reaches its peak mid-spring before decreasing until the end of summer (senescence). During autumn, grasslands located south of the Côa in the mountainous area display high vigour. RCP4.5 presents a similar pattern to historical conditions, albeit with a reduction in vigour. However, for RCP8.5, the reduction in vigour during autumn is more severe, particularly in the western regions, where the high class is mostly absent. As a result, most grasslands in Côa are expected to have low vigour under RCP8.5, except for small areas in the southeast (Penamacor). In winter, an increase in vigour is anticipated in the southern and central areas, possibly due to increased precipitation during this season and higher SPEI-1 values (Figure S3).

Spring presents a high vigour in all of the region, except in the north, where the vigour is reduced. Although projections (RCP4.5 and RCP8.5) indicate a decrease in average vigour values (Figure 7), there should be an expansion of grassland with intermediate vigour into the currently less favoured northern region. The natural growth cycle of grasslands ends in summer (growth rates near zero), with a reduction in vigour over almost the entire Côa region (2000–2021). However, under future scenarios, the vigour levels should slightly increase in the southern areas due to the effect of a higher SPEI-4 (Figure S3). Generally, while in RCP4.5, climate change is expected to cause a reduction in vigour for all Côa region to a high or low degree, the modifications would be even more drastic in RCP8.5, since in addition to causing vigour reductions, climate change is expected to affect the grassland cycle, with a direct effect on management practices.

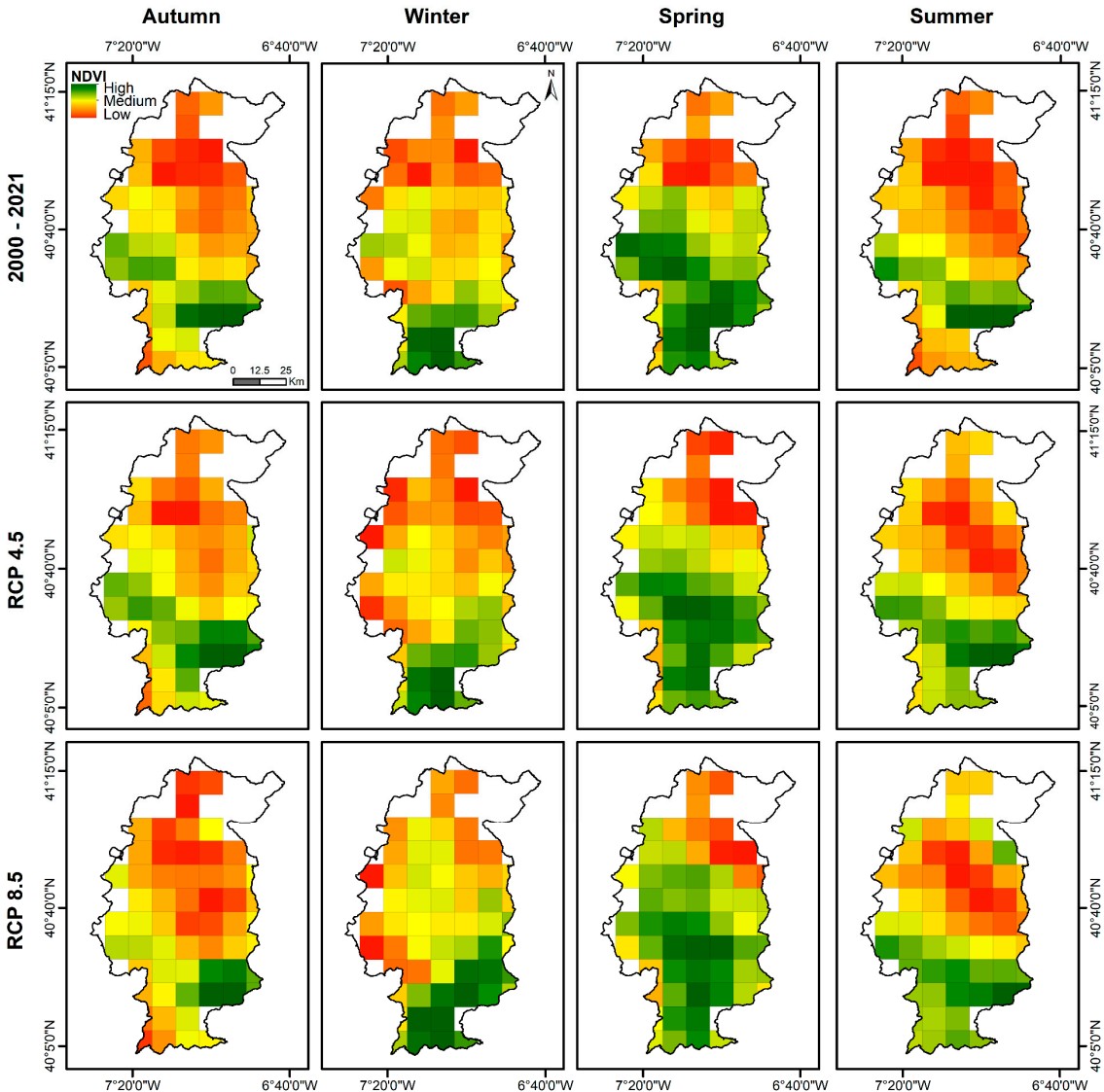

**Figure 7.** Grassland vigour in the Côa region, according to seasons for the recent past dataset (2000–2021) and the RCP4.5 and RCP8.5 scenarios.

## 4. Discussion

### 4.1. NDVI vs. SPEI

With the proposed methodology, it was possible to establish a robust relationship between SPEI, which estimates immediate and accumulated water availability, and the vegetation index (NDVI) (Figure 2b,c). A highly robust machine-learning model was then applied to estimate the non-linear spatiotemporal dynamics of grasslands under two future scenarios and generate interpretations of the real need for adaptation or mitigation measures. It was shown that SPEI-1, SPEI-4, and SPEI-12 were able to explain at least 76% of the variation in NDVI. Although other variables (e.g., temperature, precipitation, elevation, and other SPEI timesteps) may also be used, using a reduced number of predictors increases the signal-to-noise ratio, improving model robustness and avoiding overfitting. Furthermore, the SPEI already implicitly incorporates temperature and precipitation in its calculation. It is indeed considered one of the most robust drought indices, as it is obtained by the combination of precipitation and evapotranspiration over a lag period [39].

The SPEI-1 has a shorter accumulation period (1 month), thus reflecting meteorological drought and closely following the decline in soil water content [47,51]. It was the most important predictor of the model (Figure 4b) and showed that either reducing or increasing

soil moisture in a short period has a short-term response in grassland vigour, as already observed by [52]. According to [28], the SPEIs of short periods, one month or fortnight, are more suitable to provide the temporal resolution needed to improve the prediction of grassland or aboveground forage mass. Moreover, shorter time scales show large fluctuations and are more skilful in detecting more direct changes between wet and dry conditions [22].

Our study also showed that other time scales/lags may also have an impact on vigour. SPEI-4 contains the moisture memory of the previous months and reflects medium-term soil moisture conditions, providing a seasonal estimate of precipitation. The ability of seasonal medium-term SPEI values to explain annual grassland dry matter production has already been reported [29]. With SPEI-4, the effect of seasons and agricultural drought was also considered [51]. SPEI-12 provides annual drought conditions and is a good indicator of reduced soil moisture load and long-term effects on pasture plant survival. It suggests that water shortage or surplus in one year may affect soil moisture reserves in the subsequent year, with a direct effect on grassland vigour [22]. On the other hand, water shortage or surplus in one year can affect plant persistence and the future quality of the grassland. It is noted, therefore, that grasslands are very reactive to immediate meteorological events but also have long carry-over effects. Phenomena that occur over medium to long periods also affect the spectral response of grasslands. For example, one winter with little rainfall can affect grassland development in the following winter. The same can be said of droughts in one year, which can affect the water reserves in the soil as well as the density of plants covering the soil, with direct effects on the development of grasslands during the following year. It should be stated that this does not mean that immediate drought (SPEI-2 or -3) is less important for grasslands than the previous year's conditions. It is shown that all SPEIs are indeed relevant to NDVI, but their inclusion in the RF model may lead to redundancy in features as the RF model may consider the possibility that SPEI-2 or SPEI-3 might exhibit similar patterns to SPEI-1. Nonetheless, feature correlation may not always result in redundancy [53]. In order to test the abovementioned hypothesis, the RF model was re-run without SPEI-1 as input, showing that the model now selected SPEI-2 instead (Figure S4). This indeed indicates that grassland vigour is strongly associated with short-term drought, although adjacent month predictors may have redundant information for modelling.

Although the proposed methodology proved to be robust and was able to generate reliable estimates of pasture vigour, some limitations should be reported. Our approach did not explicitly consider extreme weather events, such as heatwaves, floods, snow, or frost, which are expected to become more frequent in the upcoming decades [54], nor even perturbations motivated by anthropic practices, such as fire hazards that are a relevant factor for small and isolated areas [55], which, similar to climate extremes, could cause a delay in the phenological cycle of pastures and influence the spatial distribution pattern. Furthermore, future projections of vegetation growth under different climate change scenarios may be greatly influenced by the $CO_2$ fertilisation effect [56], which was not considered in our modelling approach. This effect benefits plant growth by enhancing photosynthesis and increasing water use efficiency. Nonetheless, studies also show that this beneficial effect is indeed limited [57]. Perhaps one of the most important limitations is linked to the different grassland species' adaptation traits under future climates. In the Côa region, many different species can be found (Table S1). As such, climate change can exert varying impacts on different types of grasslands [58]. This disparity arises not only from alterations in the phenological cycle but also from changes in productivity [30]. Furthermore, climate change may lead to modifications in the dynamics of grassland ecosystems, which can ultimately result in significant alterations, e.g., to the floristic composition.

### 4.2. Impact of Climate Change on Grassland

The future increase in temperature and overall decrease in precipitation (Table 1) are expected to lead to a generalised reduction in vigour, which could drive changes in the

grassland ecosystem (Figure 5b,c). Our results show that mountainous regions should maintain their status of higher grassland vigour, although these regions will start to show lower vigour grasslands in the future [59]. There should be an overall reduction in grassland NDVI in the southern areas and a small increase in NDVI in the areas located in the north (Figure 5b,c), although not sufficient to shift to a higher vigour class. This indicates that the future reduction of precipitation in spring, summer, and autumn will be more impacting for the currently more humid areas, while the increase of precipitation in winter will be more beneficial for the currently drier areas.

The overall warmer and drier conditions favour the homogenisation of the landscape by xeric species, more resistant to heat than cold but of lower nutritional quality, especially in the case of non-managed pastures [58]. The reduction in grassland species diversity and composition are the factors that may help explain the decrease in vigour, as there is a positive and dynamic relationship between NDVI and diversity, namely in warm, dry, and antithesis periods [24]. Additionally, low diversity may increase exposed soil and moss-covered patches. These changes may force farmers to provide an additional supply of nutrients for animal feed to guarantee or maintain an adequate level of production, increasing management costs [58,60]. Nevertheless, the search for forage resources is likely to increase in mountain regions, with a potential increase in stocking density, which could increase ecosystem pressure.

Climate change in the Côa region may also influence the phenological cycle of grasslands, with effects similar to what was reported by [61], especially for colder/mountainous regions. In response to the increase in temperature, an increase in the length of the growing season of the species is expected, corroborating the results in the literature [62]. A delay in phenology is expected in autumn, which should extend into early winter. This may be related to the fact that autumn temperature increases can critically reduce water availability to plants, particularly in historically drier regions, with negative impacts on growth, photosynthesis activity and heightened risk of plant degradation and mortality [18,63]. In contrast, higher winter temperatures should favour photosynthetic growth and consumption [18], with a reduction in chlorophyll degradation and the probability of exposure to low temperatures [61].

For winter, an increase in precipitation in the Côa is also expected, which, in addition to reducing the temporary effect of water stress [24], should have a boosting effect on vigour [64], ultimately boosting pasture productivity. In our study, this is a twofold effect, mainly in the immediate drought/NDVI (SPEI-1) and the long-term drought/NDVI (SPEI-4). This helps to explain the higher NDVI in northern Côa. The peak in pasture vigour is projected to be maintained in spring but will be accompanied by a reduction in the highest values. With warming in spring, an advance in the onset of pasture flowering is also expected [59]. In practice, such conditions should favour mowing and grazing activities, which should happen earlier [18] and last until early summer. With the combined effect of reduced precipitation and higher temperatures, further degradation of grasslands in summer could be expected [22], particularly in the Côa mountainous regions. It should be noted that although drought conditions can reduce pasture yields, quality does not always decrease [65].

### 4.3. Potential Adaptation Measures for Grasslands

Some agricultural practices may be implemented to try to reduce the negative effects mentioned above. Changes in grazing periods and cutting regimes (e.g., early cuts) may dampen the negative effects of climate change on forage quality [18]. The type of grazing regime (continuous or seasonal), grazing pressure (light or heavy), and grazing season have different effects on seed bank density, with heavy grazing being unfavourable [66]. Decreasing the duration of the animal grazing sessions during the daytime and/or implementing (or increasing) night feeding sessions can be an interesting strategy to reduce the effects of moderate heat stress [18]. Additionally, controlling the stocking rate of livestock

may be able to reduce pressure on the grassland seed bank and control the consumption of plant biomass [21].

Management actions that promote increased genetic diversity are feasible options since genetic diversity allows organisms to continue adapting and evolving to new circumstances within a few generations [67]. The increase in the use of germplasm material with summer dormancy [5] is considered one of the main adaptive strategies of perennial [68] and annual grasslands [69], which allows survival through the prolonged and severe summer drought, regardless of the soil water reserve [68]. The species with higher water efficiency is indeed relevant for the future sustainable production of pasture, particularly in drylands, where reduced rainfall can limit vigour [64].

Irrigation in grasslands is also a possible intervention, although not recommended due to water scarcity. Irrigation can mitigate the effects of insufficient soil moisture, which, combined with fertilisation, may be able to promote the development of cultivated grasslands with large amounts of biomass and root accumulation in a short period [70]. Nonetheless, irrigation may be difficult to implement due to water competition (e.g., human needs and other food crops) and economic factors.

In summary, several adaptation strategies may potentially reduce the adverse effects of warming and drying projected for grasslands in the Côa region. Strategies such as the selection of varieties/species more adapted to climate change, combinations/mixtures of grassland species, changes in grazing periods, cutting regimes, grazing pressure, stocking rate, irrigation, and fertilisation are potential adaptation measures for these climate change-threatened grasslands.

## 5. Conclusions

In the present study, the developed RF model demonstrated a significant explanatory power, accounting for 76% of the variability in the NDVI. The primary drivers of grassland vigour were identified as the SPEI with temporal lags of 1, 4, and 12 months, highlighting the sensitivity of grasslands to short-term drought conditions, as well as long-term drought effects. Future projections point to a widespread reduction in grassland vigour, particularly under the RCP8.5 climate scenario. Mountainous regions, currently hosting the most vigorous grasslands, are expected to experience more pronounced negative effects, though dry lowlands in the north are anticipated to maintain their lowest vigour levels. Seasonal changes, notably a decrease in vigour during autumn, may impact grassland phenology. The results suggest that increasing temperatures and precipitation in winter may promote grassland development, which could have implications for grassland harvesting and livestock grazing timing. To safeguard vigour in less favourable zones and adapt to changing conditions, proactive adaptation measures and more efficient highland management will be essential to maintain adequate levels of grassland production. This research underscores the urgency of addressing climate change's potential impact on grasslands and offers valuable insights into the vulnerability of these ecosystems.

**Supplementary Materials:** The following supporting information can be downloaded at https://www.mdpi.com/article/10.3390/land12101914/s1, Table S1. Autochthonous plants found in the grasslands of the Côa Region. Data retrieved from the portal Flora.On (https://flora-on.pt (accessed on 4 August 2023). Figure S1. Grassland NDVI values in the Côa region for the recent past (2000–2021) and differences for two anthropogenic radiative forcing scenarios for the long-term future period (2071–2100): present (2000–2021) (a), RCP 4.5 and (b) and RCP 8.5 (c), obtained from the mean of NDVI-values. The elevation is also represented; Figure S2. Changes in grassland vigour class from the recent past (2000–2021) to the future period (2071–2100) for RCP4.5 and RCP8.5. The values are percentage representations of the number of pixels that changed or remained in the same vigour class; Figure S3. The graphs correspond to the sum of the future difference (RCP4.5 and RCP8.5, for the period 2071–2100) of the monthly values of SPEI-1, SPEI-4 and SPEI-12 relative to the current values (2000–2021). The representations were made from data obtained for two distinct regions. The left-side figures (40.1 S and −7.3 E) represent the generalized reduction of SPEI, and the right-side ones (41.2 S and −7.0 E) show a projected increase in specific months; Figure S4. Estimates of feature

importance, indicating the more relevant predictors for the RF model: (a) with all SPEIs, (b) after removing SPEI-1.

**Author Contributions:** Conceptualization, O.S. and H.F.; Data curation, O.S., A.F., C.Y. and H.F.; Formal analysis, O.S. and H.F.; Funding acquisition, H.F.; Investigation, O.S., J.A.S., H.T. and H.F.; Methodology, O.S. and H.F.; Project administration, H.F.; Resources, O.S. and H.F.; Software, O.S. and H.F.; Supervision, H.F.; Validation, O.S., J.A.S. and H.F.; Visualization, O.S. and H.F.; Writing—original draft, O.S., J.A.S., A.F., C.Y., H.T. and H.F.; Writing—review and editing, H.F. All authors have read and agreed to the published version of the manuscript.

**Funding:** This work was funded by the CoaClimateRisk project (COA/CAC/0030/2019) "O impacto das alterações climáticas e medidas de adaptação para as principais culturas agrícolas na região do Vale do Côa" financed by National Funds from the Portuguese Foundation for Science and Technology (FCT).

**Data Availability Statement:** All data used are freely available. The data that support the findings of this study are openly available in Iberia 01 (http://doi.org/10.20350/digitalCSIC/8641, accessed on 12 December 2022), E-OBS dataset (http://doi.org/10.1029/2017JD028200, accessed on 12 December 2022), Moderate Resolution Imaging Spectroradiometer (MODIS) Earth MOD09A1 (https://earthexplorer.usgs.gov/ (accessed on 12 December 2022) and at Carta de Uso e Ocupação do Solo—COS (https://snig.dgterritorio.gov.pt/, accessed on 12 December 2022).

**Acknowledgments:** We also thank projects UIDB/04033/2020, LA/P/0126/2020, and NORTE-01-0145-FEDER-000083. H.F. thanks the FCT for 2022.02317.CEECIND.

**Conflicts of Interest:** The authors declare no conflict of interest.

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
