# Peer review of "Climate Change Impacts on Grassland Vigour in Northern Portugal"

_land, doi:10.3390/land12101914_

Round 1

Reviewer 1 Report

The article entitled “Climate change impacts on grassland vigor in northern Portugal”, deals with the influence of climate change on the grasslands of northeastern Portugal. The study is well structured, the methodology used to generate the model is timely. However, the classification of grasslands as permanent and annual herbaceous is insufficient, since climate change with changes in precipitation and temperature does not affect the two large types of grasslands in the same way, since not only the phenological cycle is affected, but also its productivity, there is also a change in the dynamics of the grassland, which if this climate change persists, the grasslands change their floristic composition. I suggest the authors incorporate a classification table with the different types of grasslands, for this they can use a phytosociological classification or simply name the dominant species for each type of grass existing in the study area, for example: 1) grassland dominated by Poa bulbous. 2) Grassland dominated by Hordeum leporinum. 3) grassland dominated by woody species of……4) Grassland dominated by hemicryptophytes of……….etc.

Author Response

Reply: We would like to thank the reviewer for his/her comments, which have contributed to improve our study. The reviewer’s suggestion to incorporate a classification table with different types of grasslands based on dominant species or phytosociological characteristics, is well-considered. However, it is essential to acknowledge that, at the time of conducting this research, access to this data is very scarce. In fact, the COS landcover dataset does not include this information. Nonetheless, we have included a description of the main species that are found based on literature (regrettably very general information is available).

Line 168: “The COS dataset does not include information about species distribution, however it is known that a large number of autochthonous species can be found herein (Table S1). [37] identified the most common species in this region, including several leguminous species, such as Ornithopus compressus and Trifolium subterraneum, are found herein. Additionally, Medicago arabica is highlighted for its resilience in drier conditions. Among the grasses, Lolium multiflorum, Lolium rigidum, and Lolium perenne (ryegrasses) are prevalent, along with Dactylis glomerata and Phalaris aquatica. Various species from the genera Bromus, Vulpia, Poa, and Festuca are also significant in these areas, with Poa pratensis particularly abundant in meadows, showcasing its persistence.”

Furthermore, in light of this data limitation, we have thoughtfully added a section to our article addressing this constraint.

Line 477: “Perhaps one of the most important limitations is linked to the different grassland species adaptation traits under future climates. In the Côa region, many different species can be found (Table S1). As such, climate change can exert varying impacts on different types of grasslands [58]. This disparity arises not only from alterations in the phenological cycle but also from changes in productivity [30]. Furthermore, climate change may lead to modifications in the dynamics of grassland ecosystems, which can ultimately result in significant alterations, e.g. to the floristic composition.”

Reviewer 2 Report

Climate change is a growing challenge for the agriculture and may threaten grasslands as well. Climate change may influence the phenological cycle of grasslands. Thus stretegies needed to cope the challenge. Present paper reports the procedure for assessing the possible impacts of climate change on grasslands. Paper is written well, Introduction describes a clear background of the topic. Materials and methods are clear to understand. Prediction models are used for the purpose. Results interpreted in a good way, and discussion is written well. 

However, there must be separate heading for conclusion.

English is good, please check for spellings.

Author Response

Reply: We would like to thank the reviewer for his/her comments, which have contributed to improve our study. In particular, we have separated the discussion and the conclusion section, which now reads:

Line 550:  “5. Conclusions

In the present study, the developed RF model demonstrated a significant explanatory power, accounting for 76% of the variability in the NDVI. The primary drivers of grassland vigour were identified as the SPEI with temporal lags of 1, 4, and 12 months, highlighting the sensitivity of grasslands to short-term drought conditions, as well as long-term drought effects. Future projections point to a widespread reduction in grassland vigour, particularly under the RCP8.5 climate scenario. Mountainous regions, currently hosting the most vigorous grasslands, are expected to experience more pronounced negative effects, though dry lowlands, in the north, are anticipated to maintain their lowest vigour levels. Seasonal changes, notably a decrease in vigour during autumn, may impact grass-land phenology. The results suggests that increasing temperatures and precipitation in winter, may promote grassland development, which could have implications for the timing of grassland harvesting and livestock grazing. To safeguard vigour in less favourable zones and adapt to changing conditions, proactive adaptation measures and more efficient highland management will be essential to maintain adequate levels of grassland production. This research underscores the urgency of addressing climate change's potential impact on grasslands and offers valuable insights into the vulnerability of these eco-systems.
